palaeontology

palaeoneurobiology, Burgess Shale-type preservation, *Alalcomenaeus*, Pioche Formation, Marjum Formation

**Authors for correspondence:**
Javier Ortega-Hernández
e-mail: jortegahernandez@fas.harvard.edu
Rudy Lerosey-Aubril
e-mail: rudy_lerosey@fas.harvard.edu

# Proclivity of nervous system preservation in Cambrian Burgess Shale-type deposits

Javier Ortega-Hernández, Rudy Lerosey-Aubril and Stephen Pates

Museum of Comparative Zoology and Department of Organismic and Evolutionary Biology, Harvard University, 26 Oxford Street, Cambridge, MA 02138, USA

JO-H, 0000-0002-6801-7373; RL-A, 0000-0003-2256-1872; SP, 0000-0001-8063-9469

Recent investigations on neurological tissues preserved in Cambrian fossils have clarified the phylogenetic affinities and head segmentation in pivotal members of stem-group Euarthropoda. However, palaeoneuroanatomical features are often incomplete or described from single exceptional specimens, raising concerns about the morphological interpretation of fossilized neurological structures and their significance for early euarthropod evolution. Here, we describe the central nervous system (CNS) of the short great-appendage euarthropod *Alalcomenaeus* based on material from two Cambrian Burgess Shale-type deposits of the American Great Basin, the Pioche Formation (Stage 4) and the Marjum Formation (Drumian). The specimens reveal complementary ventral and lateral views of the CNS, preserved as a dark carbonaceous compression throughout the body. The head features a dorsal brain connected to four stalked ventral eyes, and four pairs of segmental nerves. The first to seventh trunk tergites overlie a ventral nerve cord with seven ganglia, each associated with paired sets of segmental nerve bundles. Posteriorly, the nerve cord features elongate thread-like connectives. The Great Basin fossils strengthen the original description—and broader evolutionary implications—of the CNS in *Alalcomenaeus* from the early Cambrian (Stage 3) Chengjiang deposit of South China. The spatio-temporal recurrence of fossilized neural tissues in Cambrian Konservat-Lagerstätten across North America (Pioche, Burgess Shale, Marjum) and South China (Chengjiang, Xiaoshiba) indicates that their preservation is consistent with the mechanism of Burgess Shale-type fossilization, without the need to invoke alternative taphonomic pathways or the presence of microbial biofilms.

## 1. Introduction

The Cambrian fossil record has produced fundamental insights into the morphology and initial diversification of animal phyla, with euarthropod evolution standing as a prime example of the impact of palaeontological data towards reconstructing the origin of major extant groups [1–3]. Our current understanding of the early history of euarthropods is made possible by the unusual abundance of exceptionally preserved biotas in Cambrian deposits [4–8], which capture details of the non-biomineralized anatomy that would normally be lost to decay, even under other pathways for exceptional preservation [9,10]. In addition to limbs [11,12], eyes [13,14], guts [15,16], muscles [17,18] and circulatory systems [19,20], recent studies have also reported the preservation of neurological tissues including the condensed dorsal brain, optic neuropils and the ventral nerve cord (VNC) with segmental nerves [21–30]. These discoveries directly impact hypotheses concerning the ancestral organization of the brain in extant euarthropods, and more broadly, the evolution of the head and nervous systems in Panarthropoda [31–33]. However, the preservation potential of nervous tissues in Cambrian fossils has also come under intense scrutiny, as actualistic taphonomic experiments demonstrate that the ecdysozoan nervous system is prone to rapid decay, relative to other tissues, under controlled laboratory

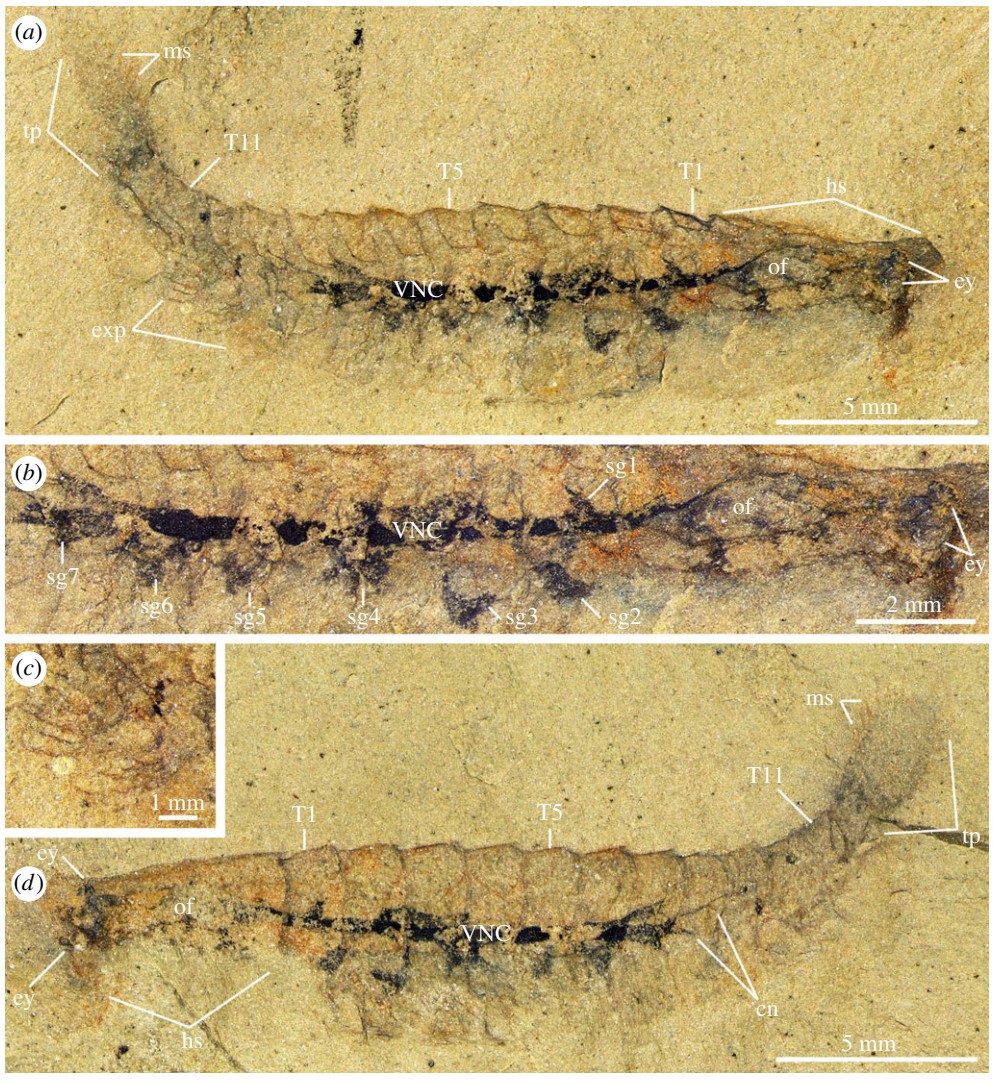

**Figure 1.** Nervous system preservation in *Alalcomenaeus* sp. from the Stage 4 Pioche Formation. (*a*) MCZ IP-197956a, part with preserved soft tissues. (*b*) Detail of CNS in the head and trunk. (*c*) Detail of trunk exopods with marginal setae. (*d*) MCZ IP-197956b, counterpart with well-preserved paddle-shaped tailspine and marginal setae. cn, connectives; ey, stalked eye; exp, trunk exopod; hs, head shield; ms, marginal setae; of, oesophageal foramen; sg*n*, segmental ganglia; T*n*, trunk tergites; VNC, ventral nerve cord. (Online version in colour.)

conditions [34,35]. Although the presence of bilateral symmetry and morphological complexity are regarded as key requisites for the valid recognition of fossilized nervous systems preserved as carbonaceous films [22–29], a reappraisal of the variability of dark organic compressions in exceptional fossils has also challenged this approach. A recent study re-interpreted putative neurological structures in the stem-group euarthropod *Fuxianhuia protensa* as microbial films that had propagated through body cavities produced by the decay of internal organs, concluding that morphology alone does not permit to unequivocally identify fossilized nervous tissues [36]. Lastly, a substantial criticism focuses on a perceived lack of reproducibility because nervous tissue preservation is extremely rare and often described from single specimens (e.g. [22,23]; but see counter-examples in [24,25,27,37,38]). Given these concerns, the input of additional fossil material from a variety of exceptional deposits could provide an effective test of the proclivity of nervous tissue preservation in Cambrian fossils. Here, we describe the nervous system in the leanchoiliid *Alalcomenaeus* from two Cambrian sites of the western USA, namely, the Stage 4 Pioche Formation and the Drumian Marjum Formation, and explore the broader implications for the preservation of neurological tissues in Burgess Shale-type deposits.

## 2. Material and methods

Studied specimens are deposited at the Harvard University Museum of Comparative Zoology (MCZ), the Kansas University Museum of Invertebrate Paleontology (KUMIP) and the Invertebrate Paleontology collections at the Smithsonian Institution (USNM). Specimens were photographed wet with crossed-polarized light using a Nikon D3X digital camera equipped with a Micro-Nikkor AF 60 mm f/2.8 D macro lens, or with normal light using a Leica IC80 HD camera mounted on a Leica M80 microscope or in a DSX110 Olympus digital microscope. Series of images were taken by manually focusing at different focal planes, and subsequently stacking and assembling in Adobe Photoshop CS6. Backscatter electron microscopy was conducted on a Quanta-650F, with a voltage of 10 kV and a working distance of 14.6 mm under low vacuum. Schematic diagrams were produced with Inkscape.

## 3. Results

### (a) *Alalcomenaeus* from the Pioche Formation

Specimen MCZ IP-197956 represents a completely articulated individual preserved in oblique-lateral view, with a total length (sag.) of 22 mm (figure 1). The dorsal exoskeleton consists of a head shield, a trunk with 11 tergites and a

tailspine. The head shield corresponds to *ca* 25% of the body length (sag.) and has an elongate isosceles-like subtrapezoidal outline with a distinctively straight anterior margin (figures 1*a*,*d* and 3*a*). The trunk includes 11 overlapping and freely articulating tergites with well-developed pleurae; MCZ IP-197956a demonstrates that the pleural tips on the left side of the body are strongly bent downwards owing to the burial orientation of the specimen (figure 1*a*). The trunk is widest (trans.) close to the head shield and tapers towards the posterior end of the body. The trunk tergites are of subequal length (sag.) on the anterior half of the body, but become slightly shorter posteriorly (figure 1*d*). Remains of the trunk appendages include traces of flat, oval-shaped exopods fringed with elongate marginal setae (figure 1*c*) and can only be directly observed in the last three trunk segments (figure 1*a*); other limbs are covered by the dorsal exoskeleton as suggested by regular impressions on the posterior trunk, or probably concealed within the rock matrix. The tailspine is paddle-shaped with a suboval outline, has a rounded posterior margin and bears acuminate setae along the margin of its distal half (figures 1*a*,*d* and 3*a*,*b*).

MCZ IP-197956 preserves various details of the internal anatomy (figures 1 and 3). The head shield covers two sets of paired ventral eyes with short but robust stalks that are connected proximally to a prominent and continuous tract-like structure that extends throughout most of the body (figure 1*a*,*b*,*d*). The morphological complexity of the tract-like structure, coupled with the regular repetition of its components and the direct connection with the eyes, suggests that it most likely represents the fossilized central nervous system (CNS) preserved as a carbonaceous film. Backscatter electron microscopy corroborates that the dark-compression tract is composed of a lighter material than the surrounding aluminosilicate matrix, as expected with Burgess Shale-type preservation [39,40] (electronic supplementary material, figure S1). There are four ventral eyes in total: two are located close to the midline and preserved in MCZ IP-197956a only, whereas the others occupy more abaxial positions as observed in MCZ IP-197956b (figures 1 and 3; electronic supplementary material, figure S2). Behind the eyes, the CNS tract splits into two branches and meets again posteriorly resulting in a prominent medial gap. This organization is consistent with the oesophageal foramen that accommodates the ventrally directed foregut and mouth opening in extant euarthropods (figure 1*b*) [22,32,41]. The presence of a well-defined gap within the dark compression also argues against its identity as remains of the gut tract, as no comparable foregut organization is known for extant or extinct euarthropods [15,16,42]. Within the trunk, the CNS tract is consolidated into a VNC that incorporates evenly spaced lateral extensions that correlate directly with the first to seventh trunk tergites, and which we interpret as the nerve bundles associated with segmental ganglia (figures 1*a*,*b*,*d* and 3*a*,*b*). Each of the first to seventh sets of nerve bundles have an irregular appearance individually and occasionally overlap each other, as observed under the first and second trunk tergites (figures 1*b* and 3*a*,*b*); such observations argue against an alternative interpretation of these structures as potential gut diverticulae, which typically exhibit a well-defined shape and a regular distribution in Cambrian euarthropods, and are particularly well known in leanchoiliids [15,16,42,43]. At the level of the seventh tergite, the VNC terminates in paired thread-like connectives that extend posteriorly (figures 1*a*,*b*,*d* and 3*a*,*b*). There is no clear indication of connectives beyond the 10th tergite, although this is probably owing to incomplete preservation rather than a legitimate biological signal.

The morphology of MCZ IP-197956 allows a confident assignment to the leanchoiliid genus *Alalcomenaeus*, previously only known from the Cambrian Stage 3 Chengjiang in South China [22] and the Wuliuan Burgess Shale in British Columbia [44]. The unique combination of characters supporting this identification includes the presence of a straight anterior head shield margin, four ventral eyes and a paddle-shaped tailspine fringed with marginal setae distally (sag.) [22,44,45]. The preserved internal anatomy of MCZ IP-197956 further strengthens this identification, as the CNS organization bears close similarities with that of *Alalcomenaeus* from Chengjiang [22] (see Discussion).

### (b) *Alalcomenaeus* from the Marjum Formation

Specimen KUMIP 204782 is an articulated individual preserved in ventral view, with a length of 60 mm (sag.) and a maximum width of 24 mm (trans.) (figures 2 and 3*c*). The dorsal exoskeleton includes the isosceles-like subtrapezoidal head shield with a straight anterior margin (figure 2*a*) and an incomplete trunk. The ventral orientation of KUMIP 204782 is evidenced by the occurrence of a small, sclerotized hypostome close to the anterior cephalic margin (figures 2*b* and 3*c*), whose posterior margin indicates the approximate position of the mouth [32]. The trunk includes at least seven freely articulating tergites of subequal length (sag.), displaying well-developed pleurae that progressively increase in curvature rearwards. The body termination is not preserved, any additional tergites are indistinctly expressed, and there are no discernible traces of the tailspine or ventral appendages.

KUMIP 204782 preserves exceptional details of the internal anatomy as a prominent tract-like dark compression that dominates the axial body region (figure 2). The tract consists of a large median band associated with pairs of segmentally arranged lateral extensions, five in the cephalic region and seven in the trunk region (figures 2 and 3*c*). The extensions composing the anteriormost pair run towards paired eyes preserved as dark ovoid compressions and thus are interpreted as optic nerves (figure 2*b*). Each optic nerve splits into two branches distally. The adaxial branches are connected with the adaxially located eyes, as observed on the right side of KUMIP 204782. The abaxial branches run towards the anterolateral corners of the hypostome, which probably conceal more medially positioned eyes (figures 2 and 3*c*). The four additional cephalic pairs of lateral extensions are interpreted as paired segmental nerve bundles that innervate the ventral appendages. Similarly, each of the seven trunk tergites is associated with a single segmental nerve pair (figures 2 and 3*c*). The interpretation of these remains as parts of the VNC is supported by the regular spacing and bilaterally symmetrical organization of the lateral extensions, their thread-like morphology and their relationships to the eyes. However, a differential coloration of some parts of the axial tract suggests that in addition to the nervous system, it also includes some remains of the gut (figures 2 and 3*c*). The gut tract displays a lighter colour and preserves some degree of three dimensionality, as revealed by low angle illumination [46], whereas the surrounding neurological tissues are preserved as dark compressions (figure 2*a*). An ovoid lighter-coloured area positioned between the optic nerves and the anteriormost

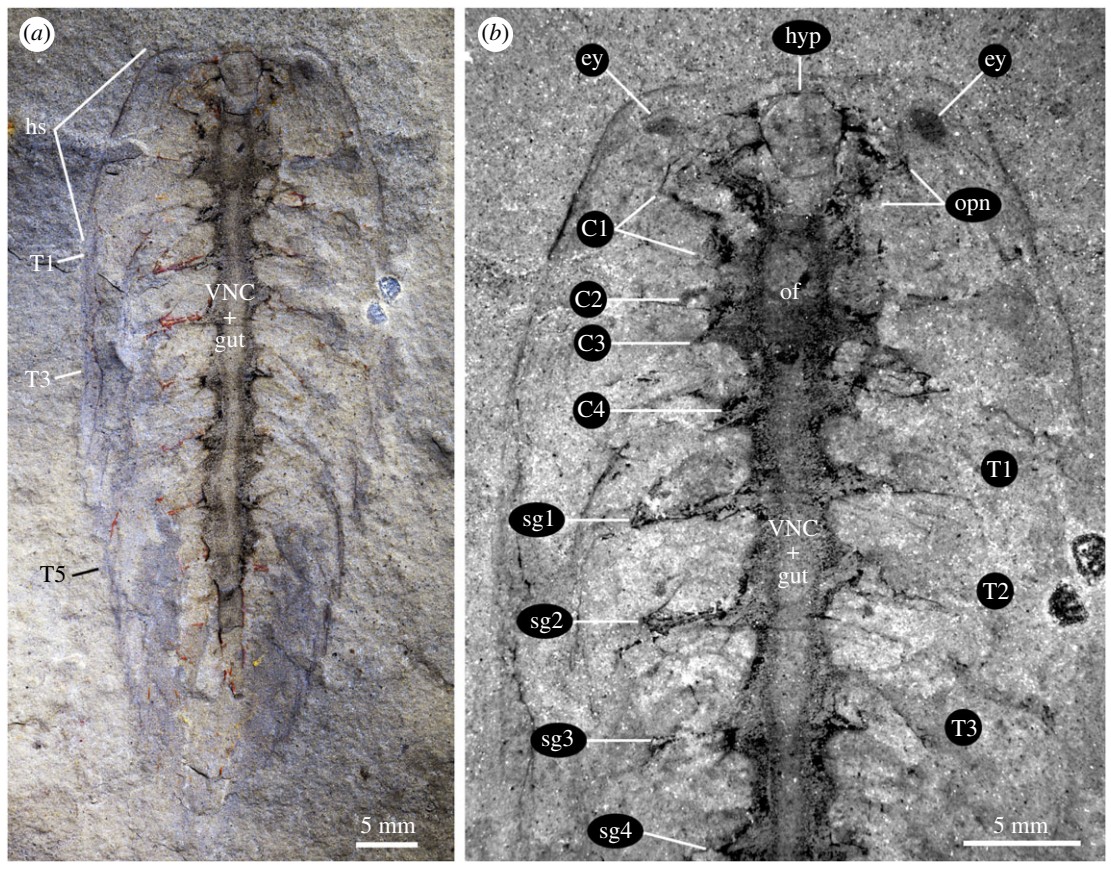

**Figure 2.** Nervous system preservation in *Alalcomenaeus* sp. from the Drumian Marjum Formation. (*a*) KUMIP 204782 photographed under cross polarized light. (*b*) Detail of anterior half photographed with ultraviolet illumination. C*n*, cephalic appendage nerves; ey, stalked eye; hyp, hypostome; hs, head shield; of, oesophageal foramen; opn, optic nerve; sg*n*, segmental ganglia; T*n*, trunk tergites.

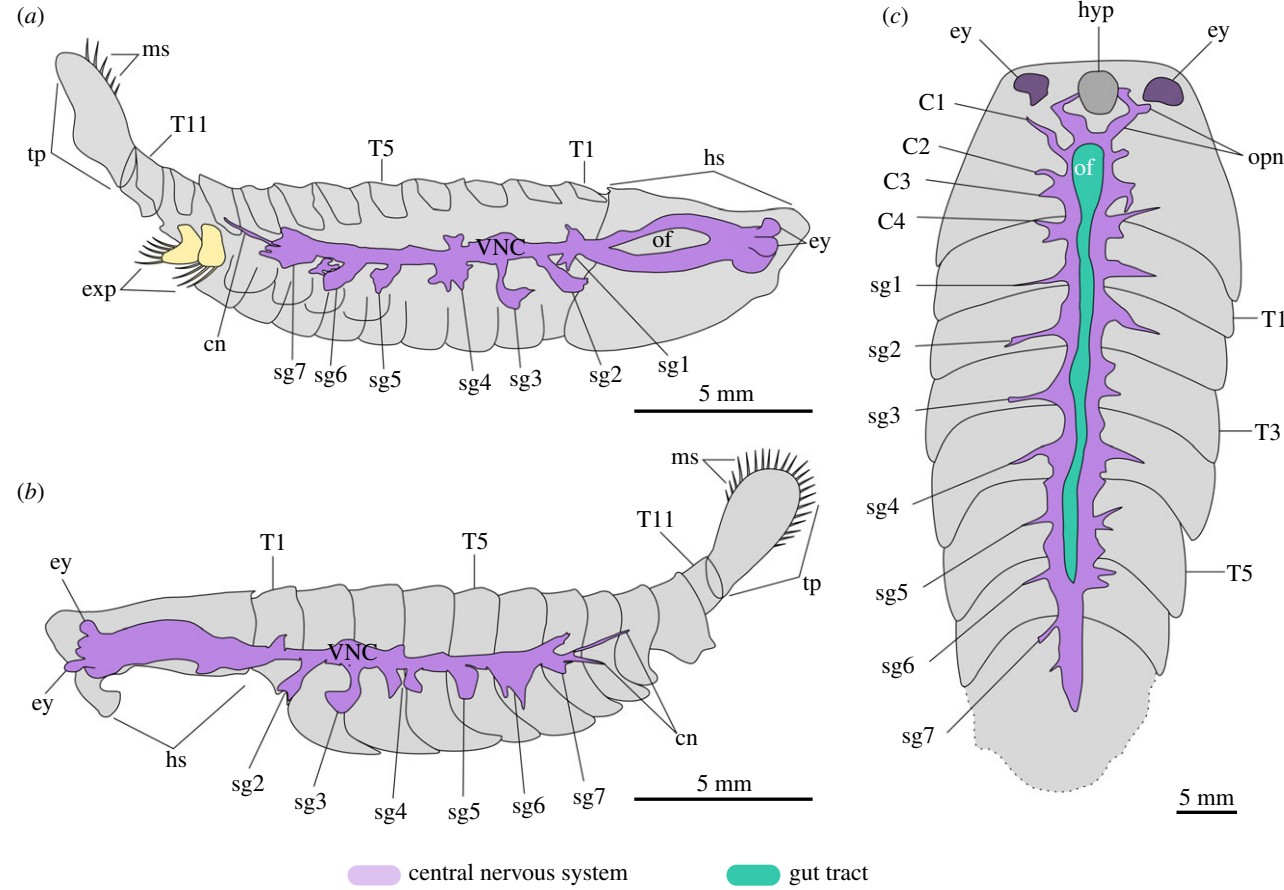

**Figure 3.** Interpretative diagram of preserved morphology in *Alalcomenaeus*. (*a*) MCZ IP-197956a part. (*b*) MCZ IP-197956b counterpart. (*c*) KUMIP 204782. C*n*, cephalic appendage nerves; cn, connectives; ey, stalked eye; exp, trunk exopod; hyp, hypostome; hs, head shield; ms, marginal setae; of, oesophageal foramen; opn, optic nerve; sg*n*, segmental ganglia; T*n*, trunk tergites; VNC, ventral nerve cord. (Online version in colour.)

three pairs of segmental nerves is tentatively regarded as the oesophageal foramen. Its location, size and outline correlate well with those of the oesophageal foramen in MCZ IP-197956 (figure 1) and *Alalcomenaeus* from Chengjiang [22].

Despite the incomplete preservation of the posterior region and the absence of appendicular data, KUMIP 204782 can be assigned to *Alalcomenaeus* based on the distinctively straight anterior margin of its subtrapezoidal head shield [44,46]. The detailed organization of the CNS further supports this interpretation [22], particularly the dichotomous optic nerves that suggest the presence of four ventral eyes, and the presence of four segmental nerve bundles in the head that would correspond to the deutocerebral and three post-oral limb pairs (figures 2*b* and 3*c*). The last major morphological revision of *Alalcomenaeus cambricus* suggested the presence of only three cephalic limb pairs, namely, the great appendages and two sets of biramous limbs [44]. However, laterally preserved *Alalcomenaeus* specimens from Chengjiang demonstrate the occurrence of an additional—but reduced—biramous limb pair behind the great appendages, similar to the closely related genus *Leanchoilia* from the Burgess Shale (see [47], figure 3*b*) and Chengjiang (see [22], figure 3; [48]). The presence of four cephalic nerve bundles behind the optic nerves in KUMIP 204782 is therefore consistent with the known appendicular cephalic organization of *Alalcomenaeus* and other members of Leanchoiliidae.

## 4. Discussion

### (a) New occurrence of *Alalcomenaeus* in North America

The new discovery of *Alalcomenaeus* from the Pioche Formation (figure 1), and confirmation of its presence in the Marjum Formation (figure 2) [46], extend the geographical distribution of this leanchoiliid genus into the western USA and stratigraphically into the Drumian. Besides the type species *A. cambricus* from the Wuliuan Burgess Shale in British Columbia [44], the genus was previously only known from the Stage 3 Chengjiang in South China. In the absence of a formal description, whether the Chinese material represents a new species remains uncertain [22]. These findings demonstrate that *Alalcomenaeus* and *Leanchoilia*, the most speciose megacheiran genus, have comparable stratigraphic and palaeogeographic ranges, the latter taxon being known from Cambrian Stage 3—Drumian strata of South China [45,48,49], Canada [47,50,51] and USA [45,52,53]. By contrast, other megacheiran genera are either endemic to their type localities (e.g. [54–57]) or are known from two different deposits at most (e.g. *Tanglangia* [58], *Yohoia* [59–61]). Finally, the presence of neurological tissues in *Alalcomenaeus* from the Marjum Formation makes this the stratigraphically youngest example to date of CNS preservation in a Cambrian Konservat-Lagerstätte (figures 4 and 5).

### (b) Consistent central nervous system morphology across space and time

The CNS of *Alalcomenaeus* from the Pioche and Marjum Formations share fundamental characteristics despite their different burial orientations in oblique and ventral views, respectively (figures 1 and 2). MCZ IP-197956 and KUMIP 204782 demonstrate the presence of four stalked ventral eyes and/or optic nerves, an oesophageal foramen and segmentally arranged nerve bundles associated with the trunk

tergites. The specimens mainly differ by virtue of their completeness. Whereas KUMIP 204782 informs the number of cephalic nerve bundles behind the eyes, these details are absent from MCZ IP-197956. Conversely, MCZ IP-197956 indicates that the VNC features consist of elongate paired connectives only in the posterior trunk region, but this part of the body is missing in KUMIP 204782. Our most significant finding is that the CNS morphology of *Alalcomenaeus* from the Pioche and Marjum Formations closely replicates the neurological organization described from Chengjiang [22] (figure 4). Complex structures including the number of ventral eyes with their respective optic nerves, oesophageal foramen, number of segmental nerve bundles in the head and trunk and the sole presence of connectives in the posterior body region are congruent between the specimens (electronic supplementary material, table S1), with the only observable discrepancies stemming from different degrees of taphonomic alteration. Our study represents, to our knowledge, the first case of consistent anatomical organization of the exceptionally preserved CNS of a Cambrian euarthropod (*Alalcomenaeus*) in three Burgess Shale-type deposits (Chengjiang, Pioche, Marjum) from two different palaeocontinents (Laurentia, South China) and three different ages (Cambrian Stage 3, Stage 4, Drumian) (figure 5; electronic supplementary material, table S1). These findings illustrate the impressive stability of the skeletal and internal anatomy of this taxon over a period of approximately 15 Myr.

### (c) Implications for taphonomy of Cambrian nervous tissues

The CNS of *Alalcomenaeus* demonstrates that Burgess Shale-type preservation can reproduce detailed and congruent neuroanatomical information in a non-biomineralizing Cambrian euarthropod, despite different physical and chemical settings in the respective palaeoenvironments at the time of deposition and subsequent taphonomic histories. The availability of three specimens of *Alalcomenaeus* indicates that preservation of the CNS in this taxon is indeed repeatable, as also exemplified by the presence of multiple individuals with nervous tissues in *Odaraia alata* [24] and *Waptia fieldensis* [28–30] from the Burgess Shale, *Fuxianhuia protensa* from Chengjiang [25], and *Chengjiangocaris kunmingensis* from the Xiaoshiba biota [27]).

The criticism of bilaterally symmetrical dark compressions exhibited by some Burgess Shale-type fossils is based on the argument that these compressions are not internal organs, but instead the remains of decay-related microbial films within body cavities [36]. The propagation of gut microbes within decaying carcasses has been thoroughly documented by laboratory experiments on the brine shrimp *Artemia* [62], which adequately explains the regularly spaced, paired subtriangular features repeatedly observed in the trunk of some Burgess Shale taxa, such as *Opabinia* [62,63], *Surusicaris* [57], *Waptia* [30] and *Yawunik* [54]. Given its rich microbiota, the digestive tract represents a focal point for the early stages of decay and the production of biofilms that, upon rupture of the gut wall, propagate within the main body and the proximal parts of the appendages [35,62]. *Opabinia* illustrates this phenomenon with exemplary clarity [64]. Well-preserved specimens display an axial dark band composed of a narrow, clearly defined, axial black stripe—most likely the gut tract replicated as a carbonaceous film—which is surrounded by a lighter-coloured material forming regularly spaced, paired

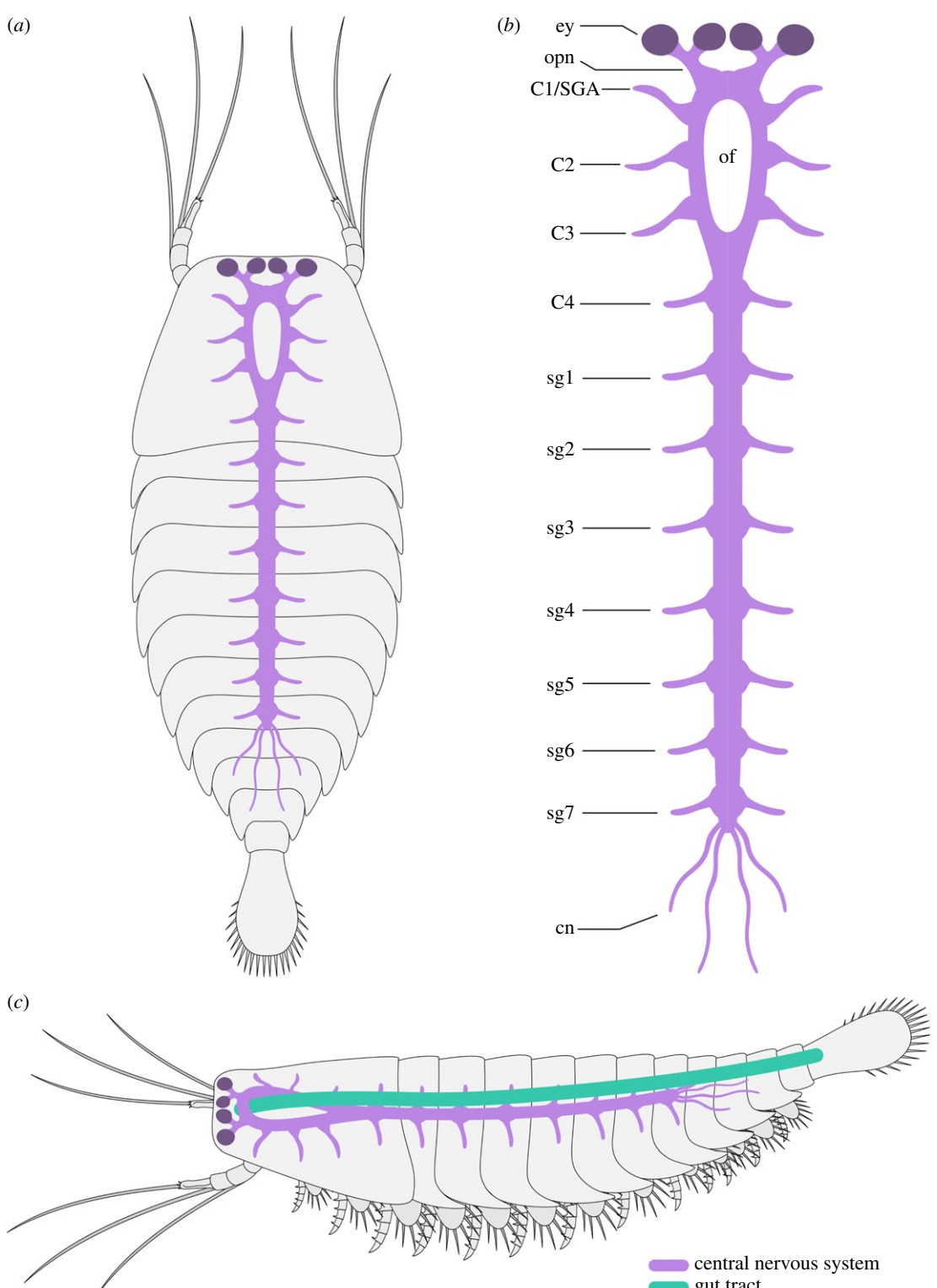

**Figure 4.** Morphological reconstruction of internal anatomy in the leanchoiliid *Alalcomenaeus*. (*a*) Dorsal view. (*b*) Organization of CNS. (*c*) Lateral view showing spatial relationship between the CNS and the overlying gut tract. Note that the morphology of the great appendages and the endopods is conjectural and based on those of *A. cambricus* [44]. cn, connectives; C*n*, cephalic appendage nerves; ey, stalked eye; of, oesophageal foramen; opn, optic nerve; SGA, short great-appendage; sg*n*, segmental ganglia. (Online version in colour.)

subtriangular lateral extensions throughout the trunk ([63,65]; figure 5). Although these lateral extensions have been interpreted as possible gut diverticulae [66], a more convincing explanation is that they are ventral limbs whose internal cavities have been partially filled with microbial films during early decay [62–64].

Although the microbial film hypothesis [36] offers a plausible alternative to the interpretation of problematic internal features observed in exceptionally preserved fossils, its ability

to closely replicate delicate and complex structures should not be overestimated. In taphonomic experiments of ecdysozoans, decay microbes propagate within the cavities of the carcass, ultimately filling most of the loosened cuticular sac [34,35,62]. The only documented mechanism in which microbes have been involved in the fine replication of a labile tissue or organ is by directly and indirectly triggering the precipitation of authigenic minerals [67–69]. The decay products observed during taphonomic experiments [34,35,62] argue

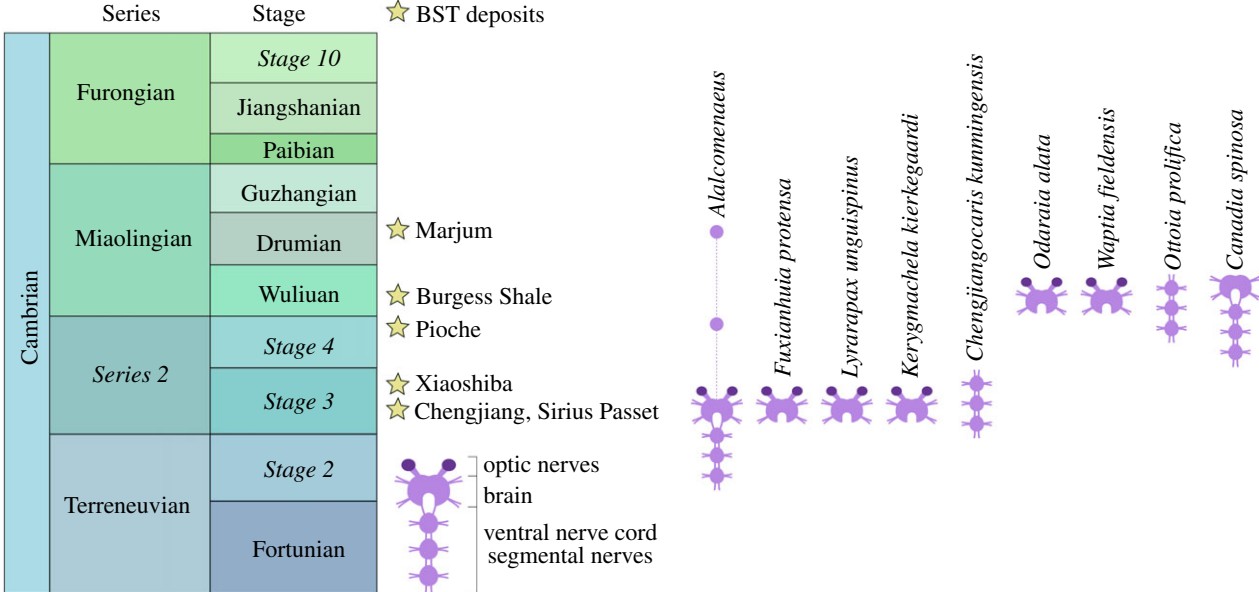

**Figure 5.** Stratigraphic distribution of Burgess Shale-type (BST) deposits with CNS preservation during the Cambrian. Note that symbols are not necessarily neuroanatomically accurate for all taxa. Based on data on the present study and [21–30,37,38]. Cases where only the optic nerves and neuropils are preserved are not included in the diagram (see the electronic supplementary material, table S1 for more information). (Online version in colour.)

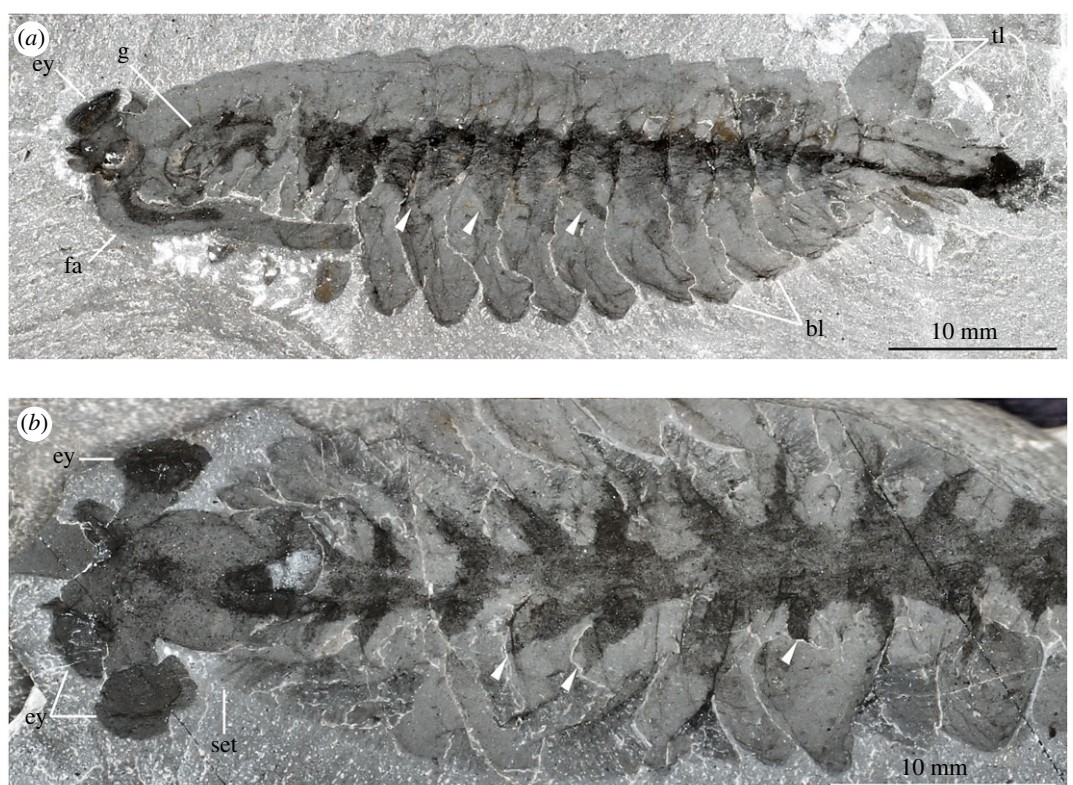

**Figure 6.** Preservation of internal structures in *Opabinia regalis* from the Wuliuan Burgess Shale. (*a*) USNM 155600, complete individual preserved in lateral view. Arrowheads point to lateral triangular extensions that most likely correspond to the lobopodous ventral limbs [62,63]. (*b*) USNM 205258, complete individual preserved in dorsal view. bl, body lobes; ey, stalked eye; fa, frontal appendage; g, gut tract; set, setae; tl, tail lobes.

against the biofilm origin of the internal structures preserved in *Alalcomenaeus*, as the latter is highly symmetrical, replicate the morphology with high fidelity and is too delicate compared to the size of the body cavity to be the result of an amorphous microbial mass. This conclusion is further strengthened when considering the spatial relationships of the internal structures relative to other anatomical features. The axial dark compression of *Alalcomenaeus* does not fill the whole cephalic region like a biofilm would be expected to do, given its

non-specific propagation. Instead, the compressions in *Alalcomenaeus* branch anteriorly to connect with the stalked eyes, which are fundamental components of the euarthropod CNS [14,32,41] and form a foramen more posteriorly that would allow the passage of the foregut (figures 1–3; electronic supplementary material, figure S2). No known internal or exoskeletal structures or cavities in the euarthropod head could have guided the propagation of microbes in a way that would fortuitously replicate the complex shape and precise

relationships of the anterior CNS. The presence of thread-like connectives in the posterior third of the trunk in two *Alalcomenaeus* specimens (figure 1*a*,*d*) [22] is also difficult to reconcile with the microbial film hypothesis for the same reason, especially because decay microbes first invade and proliferate indiscriminately in the posterior region of the body in *Artemia* [62]. Ultimately, the morphological organization, carbonaceous composition, topographical position and relationship to other body parts can be used as criteria to confidently rule out a microbial origin for organic dark compressions in Burgess Shale-type fossils and instead support their interpretation as remains of the internal anatomy.

Burgess Shale-type fossils that contain features produced by microbial activity also reveal fundamental differences with legitimate internal anatomical structures. For example, the axial dark compression of *Opabinia* is restricted to the vicinity of the gut tract and the ventral limbs and terminates at the approximate level of the mouth opening (figure 6), whereas that of *Alalcomenaeus* is directly connected with the stalked eyes anteriorly (figures 1–3; electronic supplementary material, figure S4). *Opabinia* fossils do not show a continuation of the axial tract with the anterior head [63,64,66], and similar observations also hold true for other Burgess Shale taxa with preserved guts and eyes but no direct connection between them [30,53,70]. Furthermore, *Opabinia* demonstrates that even a fossilized microbe-filled gut tract retains an important degree of morphological fidelity, for example, the presence of a J-shaped anterior curvature [63–65] (figure 6), which rules out the potential misinterpretation with neurological features such as the oesophageal foramen. These comparisons strengthen the interpretation that *Alalcomenaeus* from the Pioche and Marjum Formations, and by inference also Chengjiang [22], contain legitimate remains of the CNS whose complex morphology cannot be explained as the result of microbial activity within the body cavity (*sensu* [36,62]).

It, of course, remains of paramount importance to acknowledge the detrimental effect of decay in the quality of anatomical information available from even the most exceptional of fossils, and experimental taphonomy will continue playing a critical role in this endeavour [9,10,34,35]. However, it is also time to gravitate away from the preconception that nervous tissues are too labile to become fossilized, as evidence keeps accumulating that neurological preservation is possible through carbonaceous compressions [71], as well as authigenic mineralization [72], and even in laboratory conditions [73]. Our data demonstrate that by applying the criteria of bilateral symmetry, morphological fidelity, carbonaceous preservation, position relative to other anatomical features and congruence between multiple specimens and even localities, it is possible to identify with increasing confidence remains of the nervous system in Cambrian fossils and continue to explore their profound implications for the early evolutionary history of animals. Our results are directly relevant to euarthropod fossils, but these conclusions are also broadly applicable to other organisms from Burgess Shale-type deposits with neurological features such as the priapulid *Ottoia prolifica* [26] and the annelid *Canadia spinosa* [38]. Contrary to recent claims that the preservation of neurological tissues may require alternative taphonomic models [34,36], early and middle Cambrian Konservat-Lagerstätten from South China [21–23,25–27], North Greenland [37] and North America ([24,29,30,38], this study) consistently show CNS expressed as carbonaceous films (figure 5), in accordance with the proposed mechanism of Burgess Shale-type preservation of labile tissues [39,73]. Instead, we may seek to further refine our knowledge of how the conditions that produce Burgess Shale-type fossils contribute to the stabilization of non-biomineralizing tissues in Cambrian deposits in order to better understand the limits of exceptional preservation [74–76].

**Data accessibility.** All data are provided in the main text and electronic supplementary material.

**Authors' contributions.** J.O.-H. and R.L.-A. designed the research. J.O.-H. wrote the first draft of the manuscript and prepared the schematic diagrams, with subsequent input from R.L.-A. and S.P.; all the authors imaged and interpreted the fossil material, performed research, contributed to the discussion, and read and approved the final manuscript.

**Competing interests.** The authors declare no competing interests.

**Funding.** S.P. is funded by an Alexander Agassiz Postdoctoral Fellowship at the Museum of Comparative Zoology, Harvard University. Open access to this work is supported by a grant from the Wetmore Colles Fund.

**Acknowledgements.** Markus Martin found the specimen from the Pioche Formation, which was subsequently acquired and generously donated to the Harvard University Museum of Comparative Zoology by Andries Weug to allow its scientific study. Julien Kimmig and Bruce Lieberman facilitated the study of the specimen from the Marjum Formation. Thanks to Iris Buisman (University of Cambridge) for assistance with the use of electron microscopy facilities. We are grateful to all these people for their invaluable help during the course of our study.

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
