## [Reviewer comments · Proceedings of the Royal Society B: Biological Sciences]

Review History

RSPB-2019-2370.R0 (Original submission)

Review form: Reviewer 1

Recommendation

Accept with minor revision (please list in comments)

Scientific importance: Is the manuscript an original and important contribution to its field?

Excellent

General interest: Is the paper of sufficient general interest?

Good

Quality of the paper: Is the overall quality of the paper suitable?

Good

Is the length of the paper justified?

Yes

Should the paper be seen by a specialist statistical reviewer?

No

Do you have any concerns about statistical analyses in this paper? If so, please specify them explicitly in your report.

No

It is a condition of publication that authors make their supporting data, code and materials available - either as supplementary material or hosted in an external repository. Please rate, if applicable, the supporting data on the following criteria.

Is it accessible?

Yes

Is it clear?

Yes

Is it adequate?

Yes

Do you have any ethical concerns with this paper?

No

Comments to the Author

This study replicates interpretation of fossilised central nervous system in a Chengjiang specimen of *Alalcomenaeus* with congeners from two other Cambrian Konservat-Lagerstätten. This is an important finding, as it increases the likelihood that the structures in question are not preservational artifacts. Furthermore, the putative nervous system is preserved in carbon, as has been argued to be the case in other Cambrian biotas. I consider the authors to have a sound case that there are too many correspondences in the morphology and taphonomy of these fossils for these to be dismissed.

Page 5, line 5: I am uncertain about using “four ventral eyes” as a basis for assignment to *Alalcomenaeus*, as *Leancoilia* also has split (“4”) eyes (e.g., García-Bellido and Collins, 2007).

I recommend changing the way the posterior preserved edge of the fossil is drawn in Figure S2. By using a solid line it looks like the specimen has a rounded pygidium, where the text makes it clear that the specimen is incomplete posteriorly. Why not show the edge as broken and use the standard scarp or hatching convention to show a broken edge?

Trivial matters:

Page 1, line 30: “...data are often fragmentary”. Actually “fragmentary data” doesn’t even make much sense. Specimens are fragmentary. Data are incomplete or patchy.

Page 3, line 29: “...three trunk segments” (appendages are parts of segments rather than parts of tergites).

Page 4, line 27: “... Cambrian euarthropods, including leancoiliids”. *Leancoiliids* are almost the Poster Chilid for midgut diverticula.

Page 6, lines 12 and 23 (and page 10, line 20): “from Chengjiang” or “from the Chengjiang biota”, but not “from the Chengjiang”.

Reference 53: really incomplete reference.

Caption to Fig. S2: "Marjum" (not "Majum").

Review form: Reviewer 2 (Luke Parry)

Recommendation

Accept with minor revision (please list in comments)

Scientific importance: Is the manuscript an original and important contribution to its field?

Good

General interest: Is the paper of sufficient general interest?

Good

Quality of the paper: Is the overall quality of the paper suitable?

Excellent

Is the length of the paper justified?

Yes

Should the paper be seen by a specialist statistical reviewer?

No

Do you have any concerns about statistical analyses in this paper? If so, please specify them explicitly in your report.

No

It is a condition of publication that authors make their supporting data, code and materials available - either as supplementary material or hosted in an external repository. Please rate, if applicable, the supporting data on the following criteria.

Is it accessible?

Yes

Is it clear?

Yes

Is it adequate?

Yes

Do you have any ethical concerns with this paper?

No

Comments to the Author

Ortega Hernandez et al. report fossilized nervous tissue from specimens from the Marjum and Pioche Formations and discuss their findings in the broader context of the preservation of nervous tissue in Cambrian fossils as well as euarthropod evolution. This paper is the first report of nervous tissue from these formations and is also from sites considered to have worse

preservation overall than the Burgess and Chengjiang (Tier 2 sites sensu Gaines 2014). This reinforces the point that the preservation potential of nervous systems is not as low as over interpretation of laboratory decay experiments might have us believe. Preservation of the nervous system in the absence of the limbs is particularly intriguing regarding the relative preservation potential of limb cuticle vs nerves.

Given the rarity of these specimens and the fact that they show consistent morphology with other specimens that are >10 million years older and from different paleo regions, this paper adds important new data to the ongoing controversy on fossil nervous system preservation. This paper is also a solid review of the state of the field regarding the distribution of nervous system preservation in space and time. I recommend publication following some modest revision, mostly regarding illustration.

My primary criticism of this paper is that a lot of useful information and data is currently buried in the supplement. In order for this paper to be easily intelligible by someone who is unaccustomed to studying carbonaceous remains in BST fossils, I think that the interpretative drawings presented in figures S1 and S2 are essential. This is particularly for disentangling what is gut and what is nervous tissue in figure 2. Otherwise, the figures are nice and the images clear. Figure 5 could do with some labeling. I would not expect most readers to be able to look at photographs of *Opabinia* and be able to make sense of its anatomy relative to what is stated without it. If space is at a premium regarding moving material from the supplement to the main text, the *Opabinia* figure might be something that could be moved to the supplement.

In addition the backscatter image in figure S3b is really important, as the dark colour relative to the matrix of the material identified as nervous tissue is important as it strongly suggests carbonaceous preservation. In addition in the authors' criteria for identifying nervous tissue in BST fossils (vs dark stains for example), they should also add carbonaceous preservation, as this appears to be the pathway through which all of these features preserve across the different localities. Likewise there should be some reference to specimens from the Burgess Shale for which elemental maps have been published that show that the dark stains surrounding the body or inside heads that are visible in cross polarized light are not replicated by detectable amounts of carbon. I see lack of compositional/SEM data as being a major shortcoming of the Liu et al. paper relative to those describing nervous systems from Chengjiang fossils. Likewise it may also be worth mentioning that description of nervous tissue from non-panarthropods also strengthens the nervous system interpretation, especially in comparison to some criticisms such as those of Sansom 2016 (i.e. shrinkage during decay) and regarding microbial films infilling rigid structures. The table presented in the supplement is missing the linear dorsal structure identified as the nerve cord in *Pikaia* by Conway Morris and Caron 2012.

Minor comments:

USNM is missing from the materials and methods section for the specimen of *Opabinia*.

Line 26 – state that nervous tissue is prone to rapid decay relative to other tissues rather than just rapid decay as most non cuticularised tissues decay rapidly.

Page 9 line 9 – stage not age

Page 9 line 20 – multiple corroborating specimens are also known for *Canadia spinosa* as well as *Kerygmachela*

Review form: Reviewer 3

Recommendation

Accept with minor revision (please list in comments)

Scientific importance: Is the manuscript an original and important contribution to its field?

Excellent

General interest: Is the paper of sufficient general interest?

Excellent

Quality of the paper: Is the overall quality of the paper suitable?

Excellent

Is the length of the paper justified?

Yes

Should the paper be seen by a specialist statistical reviewer?

No

Do you have any concerns about statistical analyses in this paper? If so, please specify them explicitly in your report.

No

It is a condition of publication that authors make their supporting data, code and materials available - either as supplementary material or hosted in an external repository. Please rate, if applicable, the supporting data on the following criteria.

Is it accessible?

Yes

Is it clear?

Yes

Is it adequate?

Yes

Do you have any ethical concerns with this paper?

No

Comments to the Author

The manuscript by Ortega-Hernandez and colleagues is an outstanding contribution. The problem it addresses is an important one that is the topic of considerable current debate. The problem- do structures preserved inside many fossil arthropods represent preserved neural tissues- is a critical one that has come under recent scrutiny. The test- find and describe additional specimens from outside of South China and the Burgess Shale to determine whether the structures are consistent with neurological tissues- is sufficient and the result is a resounding yes. In my estimation, the data presented here are more than sufficient to demonstrate the preservation of nervous tissues, according to the argument beautifully laid out on p.9 (lines 5-10). This work should remove any hint of controversy moving forward.

This has application not only to the interpretation of fossils figured here, but to a host of fossils previously described from China and British Columbia. As a result, the community may now

move forward with confidence in a full understanding of the phylogenetic significance of neural tissues in Cambrian arthropods, and begin to address gaps in our understanding of extraordinary preservation by seeking to address the gap between experimental observation and the fossil record.

The manuscript is well-written, beautifully illustrated, and clear. I recommend publication with minor revision for clarity in a few places.

1. Abstract p. 3 line 8: "the diagenetic mechanism of Burgess Shale-type deposits": BST preservation is the conservation of organic remains, rather than the diagenetic alteration or replacement of tissues. Although diagenetic factors certainly aided in this, I prefer "consistent with the mechanism of Burgess Shale-type fossilization" instead, because it does not imply a diagenetic alteration of the tissues.
2. P. 3 line 15 "profound" seems out of place and unnecessary here. I suggest cutting or replacing with another term such as "current".
3. P. 3 line 17, cut "the".
4. P. 3., line 18, replace "decay under conventional taphonomic pathways" with "would normally be lost to decay, even under other pathways for exceptional preservation".
5. P. 3. line 22 "hypotheses that aim to reconstruct". I suggest replacing with "hypotheses concerning".
6. P. 4. line 4, a word is missing "morphology alone does not permit _____ to". I suggest rephrasing.
7. P. 9, line 16 "different geochemical and sedimentary context". There is no evidence presented for this, and overall, the mechanism of Burgess Shale-type preservation expects them to have been quite similar. Nevertheless, differences in depositional environment must have been present. A phrase such as "different physical and chemical settings in the respective paleoenvironments." is likely safer.
8. P. 9, line 25 "decay-induced" should be replaced with "decay-related" or simply "post-mortem". In the strict sense, decay is the loss of tissues.
9. P. 10, line 9 "filled with microbes caused by early decay" could be replaced by a phrase such as "filled with microbial films during early decay".

Overall, this is an outstanding manuscript, and I hope it will be published rapidly.

Sincerely,
Robert R. Gaines

Decision letter (RSPB-2019-2370.R0)

13-Nov-2019

Dear Mr Ortega-Hernández

I am pleased to inform you that your Review manuscript RSPB-2019-2370 entitled "Proclivity of nervous system preservation in Cambrian Burgess Shale-type deposits" has been accepted for publication in Proceedings B. Congratulations! The 3 reviewers are in remarkable agreement.

The referee(s) do not recommend any further changes. Therefore, please proof-read your manuscript carefully and upload your final files for publication. Because the schedule for publication is very tight, it is a condition of publication that you submit the revised version of your manuscript within 7 days. If you do not think you will be able to meet this date please let me know immediately.

To upload your manuscript, log into <http://mc.manuscriptcentral.com/prsb> and enter your Author Centre, where you will find your manuscript title listed under "Manuscripts with Decisions." Under "Actions," click on "Create a Revision." Your manuscript number has been appended to denote a revision.

You will be unable to make your revisions on the originally submitted version of the manuscript. Instead, upload a new version through your Author Centre.

1) A text file of the manuscript (doc, txt, rtf or tex), including the references, tables (including captions) and figure captions. Please remove any tracked changes from the text before submission. PDF files are not an accepted format for the "Main Document".

2) A separate electronic file of each figure (tiff, EPS or print-quality PDF preferred). The format should be produced directly from original creation package, or original software format. Please note that PowerPoint files are not accepted.

3) Electronic supplementary material: this should be contained in a separate file from the main text and the file name should contain the author's name and journal name, e.g. `authorname_procb_ESM_figures.pdf`

All supplementary materials accompanying an accepted article will be treated as in their final form. They will be published alongside the paper on the journal website and posted on the online figshare repository. Files on figshare will be made available approximately one week before the accompanying article so that the supplementary material can be attributed a unique DOI. Please see: <https://royalsociety.org/journals/authors/author-guidelines/>

4) Data-Sharing and data citation

It is a condition of publication that data supporting your paper are made available. Data should be made available either in the electronic supplementary material or through an appropriate repository. Details of how to access data should be included in your paper. Please see <https://royalsociety.org/journals/ethics-policies/data-sharing-mining/> for more details.

<http://datadryad.org/submit?journalID=RSPB&manu=RSPB-2019-2370> which will take you to your unique entry in the Dryad repository.

Once again, thank you for submitting your manuscript to Proceedings B and I look forward to receiving your final version. If you have any questions at all, please do not hesitate to get in touch.

Sincerely,
Professor John Hutchinson, Editor
<mailto:proceedingsb@royalsociety.org>

Associate Editor Board Member: 1

Comments to Author:

Dear Authors

You will see that the manuscript has been viewed by three referees. All believe that this study is a worthy contribution and they have made suggestions for minor revisions to be undertaken. In particular, reviewer 2 has made some suggestions to move some illustrations from the supplement to the main paper to make some of the observations clearer to the uninformed layperson.

Reviewer(s)' Comments to Author:

Referee: 1

Comments to the Author(s)

This study replicates interpretation of fossilised central nervous system in a Chengjiang specimen of *Alalcomenaeus* with congeners from two other Cambrian Konservat-Lagerstätten. This is an important finding, as it increases the likelihood that the structures in question are not preservational artifacts. Furthermore, the putative nervous system is preserved in carbon, as has been argued to be the case in other Cambrian biotas. I consider the authors to have a sound case that there are too many correspondences in the morphology and taphonomy of these fossils for these to be dismissed.

Page 5, line 5: I am uncertain about using “four ventral eyes” as a basis for assignment to *Alalcomenaeus*, as *Leaenochilia* also has split (“4”) eyes (e.g., García-Bellido and Collins, 2007).

I recommend changing the way the posterior preserved edge of the fossil is drawn in Figure S2. By using a solid line it looks like the specimen has a rounded pygidium, where the text makes it clear that the specimen is incomplete posteriorly. Why not show the edge as broken and use the standard scarp or hatching convention to show a broken edge?

Trivial matters:

Page 1, line 30: “...data are often fragmentary”. Actually “fragmentary data” doesn’t even make much sense. Specimens are fragmentary. Data are incomplete or patchy.

Page 3, line 29: “...three trunk segments” (appendages are parts of segments rather than parts of tergites).

Page 4, line 27: "... Cambrian euarthropods, including leanchoilids". Leanchoilids are almost the Poster Chilid for midgut diverticula.

Page 6, lines 12 and 23 (and page 10, line 20): "from Chengjiang" or "from the Chengjiang biota", but not "from the Chengjiang".

Reference 53: really incomplete reference.

Caption to Fig. S2: "Marjum" (not "Majum").

Referee: 2

Comments to the Author(s)

Ortega Hernandez et al. report fossilized nervous tissue from specimens from the Marjum and Pioche Formations and discuss their findings in the broader context of the preservation of nervous tissue in Cambrian fossils as well as euarthropod evolution. This paper is the first report of nervous tissue from these formations and is also from sites considered to have worse preservation overall than the Burgess and Chengjiang (Tier 2 sites sensu Gaines 2014). This reinforces the point that the preservation potential of nervous systems is not as low as over interpretation of laboratory decay experiments might have us believe. Preservation of the nervous system in the absence of the limbs is particularly intriguing regarding the relative preservation potential of limb cuticle vs nerves.

Given the rarity of these specimens and the fact that they show consistent morphology with other specimens that are >10 million years older and from different paleo regions, this paper adds important new data to the ongoing controversy on fossil nervous system preservation. This paper is also a solid review of the state of the field regarding the distribution of nervous system preservation in space and time. I recommend publication following some modest revision, mostly regarding illustration.

My primary criticism of this paper is that a lot of useful information and data is currently buried in the supplement. In order for this paper to be easily intelligible by someone who is unaccustomed to studying carbonaceous remains in BST fossils, I think that the interpretative drawings presented in figures S1 and S2 are essential. This is particularly for disentangling what is gut and what is nervous tissue in figure 2. Otherwise, the figures are nice and the images clear. Figure 5 could do with some labeling. I would not expect most readers to be able to look at photographs of Opabinia and be able to make sense of its anatomy relative to what is stated without it. If space is at a premium regarding moving material from the supplement to the main text, the Opabinia figure might be something that could be moved to the supplement.

In addition the backscatter image in figure S3b is really important, as the dark colour relative to the matrix of the material identified as nervous tissue is important as it strongly suggests carbonaceous preservation. In addition in the authors' criteria for identifying nervous tissue in BST fossils (vs dark stains for example), they should also add carbonaceous preservation, as this appears to be the pathway through which all of these features preserve across the different localities. Likewise there should be some reference to specimens from the Burgess Shale for which elemental maps have been published that show that the dark stains surrounding the body or inside heads that are visible in cross polarized light are not replicated by detectable amounts of carbon. I see lack of compositional/SEM data as being a major shortcoming of the Liu et al. paper relative to those describing nervous systems from Chengjiang fossils. Likewise it may also be

worth mentioning that description of nervous tissue from non-panarthropods also strengthens the nervous system interpretation, especially in comparison to some criticisms such as those of Sansom 2016 (i.e. shrinkage during decay) and regarding microbial films infilling rigid structures. The table presented in the supplement is missing the linear dorsal structure identified as the nerve cord in *Pikaia* by Conway Morris and Caron 2012.

Minor comments:

USNM is missing from the materials and methods section for the specimen of *Opabinia*.

Line 26 – state that nervous tissue is prone to rapid decay relative to other tissues rather than just rapid decay as most non cuticularised tissues decay rapidly.

Page 9 line 9 – stage not age

Page 9 line 20 – multiple corroborating specimens are also known for *Canadia spinosa* as well as *Kerygmachela*

Referee: 3

Comments to the Author(s)

The manuscript by Ortega-Hernandez and colleagues is an outstanding contribution. The problem it addresses is an important one that is the topic of considerable current debate. The problem- do structures preserved inside many fossil arthropods represent preserved neural tissues- is a critical one that has come under recent scrutiny. The test- find and describe additional specimens from outside of South China and the Burgess Shale to determine whether the structures are consistent with neurological tissues- is sufficient and the result is a resounding yes. In my estimation, the data presented here are more than sufficient to demonstrate the preservation of nervous tissues, according to the argument beautifully laid out on p.9 (lines 5-10). This work should remove any hint of controversy moving forward.

This has application not only to the interpretation of fossils figured here, but to a host of fossils previously described from China and British Columbia. As a result, the community may now move forward with confidence in a full understanding of the phylogenetic significance of neural tissues in Cambrian arthropods, and begin to address gaps in our understanding of extraordinary preservation by seeking to address the gap between experimental observation and the fossil record.

The manuscript is well-written, beautifully illustrated, and clear. I recommend publication with minor revision for clarity in a few places.

1. Abstract p. 3 line 8: "the diagenetic mechanism of Burgess Shale-type deposits": BST preservation is the conservation of organic remains, rather than the diagenetic alteration or replacement of tissues. Although diagenetic factors certainly aided in this, I prefer "consistent with the mechanism of Burgess Shale-type fossilization" instead, because it does not imply a diagenetic alteration of the tissues.

2. P. 3 line 15 "profound" seems out of place and unnecessary here. I suggest cutting or replacing with another term such as "current".

3. P. 3 line 17, cut "the".

4. P. 3., line 18, replace "decay under conventional taphonomic pathways" with "would normally be lost to decay, even under other pathways for exceptional preservation".

5. P. 3. line 22 "hypotheses that aim to reconstruct". I suggest replacing with "hypotheses concerning".

6. P. 4. line 4, a word is missing "morphology alone does not permit _____ to". I suggest rephrasing.

7. P. 9, line 16 "different geochemical and sedimentary context". There is no evidence presented for this, and overall, the mechanism of Burgess Shale-type preservation expects them to have been quite similar. Nevertheless, differences in depositional environment must have been present. A phrase such as "different physical and chemical settings in the respective paleoenvironments." is likely safer.

8. P. 9, line 25 "decay-induced" should be replaced with "decay-related" or simply "post-mortem". In the strict sense, decay is the loss of tissues.

9. P. 10, line 9 "filled with microbes caused by early decay" could be replaced by a phrase such as "filled with microbial films during early decay".

Overall, this is an outstanding manuscript, and I hope it will be published rapidly.

Sincerely,
Robert R. Gaines

Author's Response to Decision Letter for (RSPB-2019-2370.R0)

See Appendix A.

Decision letter (RSPB-2019-2370.R1)

15-Nov-2019

Dear Dr Ortega-Hernández

I am pleased to inform you that your manuscript entitled "Proclivity of nervous system preservation in Cambrian Burgess Shale-type deposits" has been accepted for publication in Proceedings B.

Open Access

Paper charges

Sincerely,

Appendix A

Associate Editor Board Member: 1

Comments to Author:

Dear Authors

You will see that the manuscript has been viewed by three referees. All believe that this study is a worthy contribution and they have made suggestions for minor revisions to be undertaken. In particular, reviewer 2 has made some suggestions to move some illustrations from the supplement to the main paper to make some of the observations clearer to the uninformed layperson.

R. We greatly appreciate the positive feedback on this contribution! We have followed all the recommendations below and incorporated some of the supplementary figures into the main text as per the reviewer's suggestions.

Reviewer(s)' Comments to Author:

Referee: 1

Comments to the Author(s)

This study replicates interpretation of fossilised central nervous system in a Chengjiang specimen of *Alalcomenaeus* with congeners from two other Cambrian Konservat-Lagerstätten. This is an important finding, as it increases the likelihood that the structures in question are not preservational artifacts. Furthermore, the putative nervous system is preserved in carbon, as has been argued to be the case in other Cambrian biotas. I consider the authors to have a sound case that there are too many correspondences in the morphology and taphonomy of these fossils for these to be dismissed.

R. Thanks for the positive commentary!

Page 5, line 5: I am uncertain about using “four ventral eyes” as a basis for assignment to *Alalcomenaeus*, as *Leanchoilia* also has split (“4”) eyes (e.g., García-Bellido and Collins, 2007).

R. The presence of four ventral eyes represents one of the morphological characters that support the assignment of specimen MCZ IP-197956 to *Alalcomenaeus*, but which also include the anterior straight cephalic margin and a paddle shaped tailspine with marginal spines. As correctly pointed out by the reviewer the condition of the eyes is not exclusive to *Alalcomenaeus*, but does indeed represent an important character supporting this interpretation based on the results of Tanaka et al. (2013 *Nature*) from Chengjiang. We have clarified this point by slightly rephrasing this sentence for clarity.

I recommend changing the way the posterior preserved edge of the fossil is drawn in Figure S2. By using a solid line it looks like the specimen has a rounded pygidium, where the text makes it clear that the specimen is incomplete posteriorly. Why not show the edge as broken and use the standard scarp or hatching convention to show a broken edge?

R. Done.

Trivial matters:

Page 1, line 30: "...data are often fragmentary". Actually "fragmentary data" doesn't even make much sense. Specimens are fragmentary. Data are incomplete or patchy.

R. We have changed "fragmentary" to "incomplete".

Page 3, line 29: "...three trunk segments" (appendages are parts of segments rather than parts of tergites).

R. Change done.

Page 4, line 27: "... Cambrian euarthropods, including leanchoiliids". Leanchoiliids are almost the Poster Chilid for midgut diverticula.

R. We have expanded this sentence to emphasize the significance of leanchoiliids in this context and included a reference to the seminal publication by Butterfield (2002 *Paleobiology*) on this topic. Consequently, we have also renumbered the remaining references.

Page 6, lines 12 and 23 (and page 10, line 20): "from Chengjiang" or "from the Chengjiang biota", but not "from the Chengjiang".

R. Change done throughout the manuscript.

Reference 53: really incomplete reference.

R. Change done.

Caption to Fig. S2: "Marjum" (not "Majum").

R. Change done.

Referee: 2

Comments to the Author(s)

Ortega Hernandez et al. report fossilized nervous tissue from specimens from the Marjum and Pioche Formations and discuss their findings in the broader context of the preservation of nervous tissue in Cambrian fossils as well as euarthropod evolution. This paper is the first report of nervous tissue from these formations and is also from sites considered to have worse preservation overall than the Burgess and Chengjiang (Tier 2 sites sensu Gaines 2014). This reinforces the point that the preservation potential of nervous systems is not as low as over interpretation of laboratory decay experiments might have us believe. Preservation of the nervous system in the absence of the limbs is particularly intriguing regarding the relative preservation potential of limb cuticle vs nerves. Given the rarity of these specimens and the fact that they show consistent morphology with other specimens that are >10 million years older and from different paleo regions, this paper adds important new data to the ongoing controversy on fossil nervous system preservation. This paper is also a solid review of the state of the field regarding the distribution of nervous system preservation in space and time. I recommend publication following some modest revision, mostly regarding illustration.

R. Thanks for the positive commentary!

My primary criticism of this paper is that a lot of useful information and data is currently buried in the supplement. In order for this paper to be easily intelligible by someone who is unaccustomed to studying carbonaceous remains in BST fossils, I think that the interpretative drawings presented in figures S1 and S2 are essential. This is particularly for disentangling what is gut and what is nervous tissue in figure 2.

R. Done. We have combined figures S1 and S2 into a single figure in the main text, the new Figure 3, and have modified the remaining illustrations accordingly.

Otherwise, the figures are nice and the images clear. Figure 5 could do with some labeling. I would not expect most readers to be able to look at photographs of Opabinia and be able to make sense of its anatomy relative to what is stated without it. If space is at a premium regarding moving material from the supplement to the main text, the Opabinia figure might be something that could be moved to the supplement.

R. Done.

In addition the backscatter image in figure S3b is really important, as the dark colour relative to the matrix of the material identified as nervous tissue is important as it strongly suggests carbonaceous preservation. In addition in the authors' criteria for identifying nervous tissue in BST fossils (vs dark stains for example), they should also add carbonaceous preservation, as this appears to be the pathway through which all of these features preserve across the different localities.

R. Done. We have included additional discussions on the significance of confirmed carbonaceous preservation for the recognition of neurological features in Burgess Shale type fossils.

Likewise there should be some reference to specimens from the Burgess Shale for which elemental maps have been published that show that the dark stains surrounding the body or inside heads that are visible in cross polarized light are not replicated by detectable amounts of carbon. I see lack of compositional/SEM data as being a major shortcoming of the Liu et al. paper relative to those describing nervous systems from Chengjiang fossils.

R. Done, we have included new information on the Table S1 to reflect these details.

Likewise it may also be worth mentioning that description of nervous tissue from non-panarthropods also strengthens the nervous system interpretation, especially in comparison to some criticisms such as those of Sansom 2016 (i.e. shrinkage during decay) and regarding microbial films infilling rigid structures.

R. Change done. We have included mention of non-euarthropods in the discussion in support of the presence of neurological structures in Burgess Shale type deposits.

The table presented in the supplement is missing the linear dorsal structure identified as the nerve cord in Pikaia by Conway Morris and Caron 2012.

R. Change done.

Minor comments:

USNM is missing from the materials and methods section for the specimen of Opabinia.

R. Change done.

Line 26 – state that nervous tissue is prone to rapid decay relative to other tissues rather than just rapid decay as most non cuticularised tissues decay rapidly.

R. Change done.

Page 9 line 9 – stage not age

R. Change done.

Page 9 line 20 – multiple corroborating specimens are also known for *Canadia spinosa* as well as *Kerygmachela*

R. We have included mentions to these references earlier in the manuscript for clarity.

Referee: 3

Comments to the Author(s)

The manuscript by Ortega-Hernandez and colleagues is an outstanding contribution. The problem it addresses is an important one that is the topic of considerable current debate. The problem- do structures preserved inside many fossil arthropods represent preserved neural tissues- is a critical one that has come under recent scrutiny. The test- find and describe additional specimens from outside of South China and the Burgess Shale to determine whether the structures are consistent with neurological tissues- is sufficient and the result is a resounding yes. In my estimation, the data presented here are more than sufficient to demonstrate the preservation of nervous tissues, according to the argument beautifully laid out on p.9 (lines 5-10). This work should remove any hint of controversy moving forward.

R. Thanks for the positive commentary!

This has application not only to the interpretation of fossils figured here, but to a host of fossils previously described from China and British Columbia. As a result, the community may now move forward with confidence in a full understanding of the phylogenetic significance of neural tissues in Cambrian arthropods, and begin to address gaps in our understanding of extraordinary preservation by seeking to address the gap between experimental observation and the fossil record.

The manuscript is well-written, beautifully illustrated, and clear. I recommend publication with minor revision for clarity in a few places.

1. Abstract p. 3 line 8: "the diagenetic mechanism of Burgess Shale-type deposits": BST preservation is the conservation of organic remains, rather than the diagenetic alteration or replacement of tissues. Although diagenetic factors certainly aided in this, I prefer "consistent with the mechanism of Burgess Shale-type fossilization" instead, because it does not imply a diagenetic alteration of the tissues.

R. Change done.

2. P. 3 line 15 "profound" seems out of place and unnecessary here. I suggest cutting or replacing with another term such as "current".

R. Change done.

3. P. 3 line 17, cut "the".

R. Change done.

4. P. 3., line 18, replace "decay under conventional taphonomic pathways" with "would normally be lost to decay, even under other pathways for exceptional preservation".

R. Change done.

5. P. 3. line 22 "hypotheses that aim to reconstruct". I suggest replacing with "hypotheses concerning".

R. Change done.

6. P. 4. line 4, a word is missing "morphology alone does not permit _____ to". I suggest rephrasing.

R. Change done.

7. P. 9, line 16 "different geochemical and sedimentary context". There is no evidence presented for this, and overall, the mechanism of Burgess Shale-type preservation expects them to have been quite similar. Nevertheless, differences in depositional environment must have been present. A phrase such as "different physical and chemical settings in the respective paleoenvironments." is likely safer.

R. Change done.

8. P. 9, line 25 "decay-induced" should be replaced with "decay-related" or simply "post-mortem". In the strict sense, decay is the loss of tissues.

R. Change done.

9. P. 10, line 9 " filled with microbes caused by early decay" could be replaced by a phrase such as "filled with microbial films during early decay".

R. Change done.

Overall, this is an outstanding manuscript, and I hope it will be published rapidly.

Sincerely,
Robert R. Gaines